# Preparation of Novel Mesoporous LaFeO_3_-SBA-15-CTA Support for Syngas Formation of Dry Reforming

**DOI:** 10.3390/nano12091451

**Published:** 2022-04-24

**Authors:** Luming Li, Song Wu, Hongmei Li, Jie Deng, Junshan Li

**Affiliations:** 1College of Food and Biological Engineering, Chengdu University, Chengdu 610106, China; liluming@cdu.edu.cn (L.L.); wusong@stu.cdu.edu.cn (S.W.); lihongmei@cdu.edu.cn (H.L.); 2Institute for Advanced Study, Chengdu University, Chengdu 610106, China; 3Department of Chemical Engineering, Sichuan University, Chengdu 610065, China

**Keywords:** LaFeO3-SBA-15-CTA, Ni, dry reforming, syngas, interaction

## Abstract

A nanocomposite NiPt/5LSBA-160 catalyst comprised of highly dispersed Ni nanoparticles contacting intimately with Pt over novel mesoporous LaFeO_3_-SBA-15-CTA support with a high specific surface area (SSA) was successfully developed for the dry reforming of methane. Results revealed that the high SSA mesoporous LaFeO_3_-SBA-15-CTA materials could first be synthesized by an in situ growth hydrothermal process and used as an excellent carrier candidate of Ni-based catalysts to achieve enhanced catalytic activity due to the strong interaction between LaFeO_3_ and Ni species. Moreover, the introduction of Pt over a Ni/5LSBA-160 catalyst would further promote the interaction between Ni and support, improve the dispersion of active Ni centers and obtain a higher syngas formation rate as well as tolerance to carbon coking than that of a Pt-free Ni/5LSBA-160 catalyst sample. This finding uncovers a promising prospect for high SSA mesoporous perovskite preparation and utilization in catalysis such as oxidation, hydrogenation, photocatalysis, energy conversion and so on.

## 1. Introduction

The catalytic dry reforming of methane (DRM) process can convert both CH_4_ and CO_2_ into synthesis gas (syngas, CO and H_2_), which is a classical building feed to produce sulfur-free diesel fuels, oxygenated chemicals and other important high value-added chemicals [1,2,3]. However, the DRM reaction usually needs to be performed at a high reaction temperature due to its intrinsic endothermic property (Equation (1)), which inevitably generates some side reactions, such as Equations (2) and (3), and causes carbon deposition, covering the active site of the catalysts [4,5,6]. On the other hand, the catalysts (e.g., Ni-based catalysts) universally exhibit low thermal stability under harsh reaction conditions which leads to sintering [7]. Therefore, to realize the industrialization of the DRM reaction with a long lifetime and high efficiency, two major manipulators for catalysts must first be solved, namely active phase (Ni) sintering and carbon deposition [8,9].


(1)
$${\mathrm{CH}}_{4}+{\mathrm{CO}}_{2}⇌2H_{2}+2\mathrm{CO}\;△H_{298}^{\theta }=247\;\mathrm{KJ}/\mathrm{mol}$$



(2)
$$2\mathrm{CO}⇌C+{\mathrm{CO}}_{2}\;△H_{298}^{\theta }=-172\;\mathrm{KJ}/\mathrm{mol}$$



(3)
$$2\mathrm{CO}⇌C+{\mathrm{CO}}_{2}\;△H_{298}^{\theta }=-172\;\mathrm{KJ}/\mathrm{mol}$$


It is accepted that the strong metal–support interaction can serve as an alternative effective strategy to improve the dispersion of active phase nanoparticles and promote their resistance to sintering and carbon deposition [10,11]. The key issue is to develop carriers or additives with high thermal stability that can strongly interact with the active phase. Various materials such as Al_2_O_3_, ZrO_2_, MgO, CeO_2_, mesoporous silica and molecular sieve are used as catalyst supports or additives and can enhance the catalytic activity of active species such as Ni particles in DRM [12]. Recently, perovskite oxides (ABO_3_) aroused much attention due to their outstanding virtues of abundant lattice defects, readily heteroatomic substitution and high thermal stability for the DRM reaction [13,14]. Catherine et al. reported that an LaNiO_3_ catalyst prepared using the auto-ignition process showed high activity, stability and good tolerance to anti-coke for dry reforming [15]. Moradi et al. studied the partial substitution of Mg, Sr and Ba on the A site and evaluated the catalytic activity for DRM. It was found that the substitution of Ba showed higher activity than that of Mg and Sr, and the optimized La_0.9_Ba_0.1_NiO_3_ catalyst possessed the highest conversion of CH_4_ and CO_2_. Moreover, LaFeO_3_-based catalysts also performed high catalytic activity and the selectivity of synthesis gas (CO and H_2_) [16,17]. Liao et al. reported that the Ni nanoparticles were served as catalytically active sites self-generated from nanoparticles/LaFeO_3_ heterogeneous structure, and can exhibit superior performance for both methane conversion and the activation of C–O bonds [18].

Nevertheless, the preparation of perovskites should be performed under relatively stringent conditions such as high temperature and long calcinations time; this would cause the agglomeration of grain size and decrease of specific surface area, which would discourage the exposure of active sites or loading center for heterogeneous catalysis [19]. To deal with this problem, some porous silica materials such as ordered cubic mesopores (KIT-6) and ordered hexagonal mesoporous silica structures (SBA-15) were added during the synthesis process as hard templates [20]. Mesoporous LaFeO_3_ catalyst was successfully formed using KIT-6 as a hard template, and the specific surface area was up to138 m^2^/g, which exhibited a higher adsorbed oxygen concentration and better low-temperature reducibility as well as excellent oxidation activity [21]. Ruan et al. employed SBA-15 as a templating agent to prepare a mesoporous LaAl_0.25_Ni_0.75_O_3_ perovskite catalyst, and the catalyst process had higher activity and stability than that of the catalyst synthesized by the commercial silica template, where the conversion for both CH_4_ and CO_2_ can be maintained at more than 75% after 36 h of DRM reaction [20]. However, the subsequent removal process of the templating agent is complicated as well as the discharge of wastewater. Therefore, it is urgent to develop a new synthesis process to prepare perovskites-based catalysts with large specific surface area and multiple active catalytic sites for DRM reaction.

In our previous study, high-quality SBA-15-CTA (CTA stands for citric acid) materials were tailored under moderate polycarboxylic acid (citric acid), and the obtained SBA-15-CTA-loaded Ni catalysts showed good potential for the dry reforming of methane (DRM) reaction with much less coke formation at 700 °C [22]. Moreover, citric acid is widely used as a complexing agent to promote intermetallic dispersion and reduce the crystallization temperature in the preparation of the tunable perovskite materials [23]. The bridging connection of citric acid spurs us to develop an SBA-15-CTA supported perovskites catalyst with a large specific surface area using one-pot hydrothermal technology. In this contribution, a one-step in situ growth hydrothermal process was firstly employed to effectively prepare a series of LaFeO_3_/SBA-15-CTA hybrid materials with a large specific surface area, and these mesoporous perovskites were used as the carriers to synthesize Ni(Pt)/LaFeO_3_-SBA hybrid catalysts for the DRM reaction. The obtained composite catalysts displayed high activity and resistance to carbon deposition for the dry reforming process. It should be noted that the preliminary study for the addition of Pt aims at investigating the interaction among the metals and the support, and further investigations are needed.

## 2. Materials and Methods

### 2.1. Materials

All the reagents, including anhydrous citric acid, tetraethyl orthosilicate (≥98%, TEOS), poly (ethylene glycol)-block-poly (propylene glycol)-block-poly (ethylene glycol) (P_123_), lanthanum nitrate hexahydrate (La(NO_3_)_3_·6H_2_O), ferric nitrate nonahydrate (Fe(NO_3_)_3_·9H_2_O), chloroplatinic acid hexahydrate (H_2_PtCl_6_·6H_2_O) and nickel nitrate hexahydrate (Ni(NO_3_)_2_·6H_2_O) were provided by Aldrich. It is worthy to point out that all the chemicals were of analytical grade and carried out without any pretreatment.

### 2.2. Catalysts Preparation

#### 2.2.1. Preparation of LaFeO_3_-SBA-15-CTA Supports

The preparation of LaFeO_3_-SBA-15-CTA supports was carried out using a one-pot in situ growth hydrothermal process. The precursor solution was produced by mixing stoichiometric amounts of La(NO_3_)_3_·6H_2_O and Fe(NO_3_)_3_·9H_2_O into citric acid-deionized water solution (citric/(La + Fe) molar ratio = 1.1) at room temperature (RT), and then the surfactant of P_123_ was added to the solution and stirred to form a homogeneous solution. After that, TEOS was slowly added to the above mixture. Subsequently, the mixture was loaded into a PTFE-lined stainless-steel autoclave and heated at various temperatures (120 °C, 140 °C, 160 °C) for 24 h (please refer to Table 1 for the detailed experimental factors). After cooling down to room temperature, the crystallized product was filtered, washed and dried at 100 °C for 24 h. Finally, the powder product was calcined at 750 °C for 6 h to achieve mesoporous xLaFeO_3_-SBA-15-CTA support, which was labeled as xLSBA-T (x means the content of LaFeO_3_, T stands for the hydrothermal temperature).

#### 2.2.2. Preparation of NiO(PtO_x_)/yLSBA-T Catalysts

NiO(PtO_x_)/yLSBA-T catalysts were prepared via the wet impregnation method using an aqueous solution containing Ni(NO_3_)_2_·6H_2_O and H_2_PtCl_6_·6H_2_O. All impregnated samples were sequentially stirred, rotary evaporated, dried and calcinated at 750 °C for 6 h, as in our previous report [24]. It was noted that the Ni and Pt components were settled at 5 wt.% and 0.5 wt.%, respectively.

### 2.3. Catalysts Characterization

X-ray diffraction (XRD, 2θ = 5–80°) and small-angle XRD (2θ = 0–6°) were carried out on an X-ray diffractometer (Bruker D8 Advance) using Cu Kα1 irradiation (λ = 0.15418 nm). N_2_ adsorption–desorption experiments were performed on a Micromeritics ASAP 2420 automatic analyzer to achieve the specific surface area and pore volume. Prior to testing, the samples were pretreated at 200 °C for 12 h to remove impurities. The microstructure of samples was analyzed via scanning electron microscopy (SEM, JEOL, JEM-2100F, Tokyo, Japan) and transmission electron microscopy (TEM, FEI Tecnai G2 F20, Waltham, MA, USA) at 200 kV. An AutoChem II 2920 (Thermo Scientific, Waltham, MA, USA) was engaged to study the reduction behavior of the NiO(PtO_x_)/yLSBA-T catalysts. A 30 mg sample was used, which was pretreated under Ar flow (30 mL/min) at 300 °C for 1 h. After that, the sample was heated in a 30 mL/min 10% H_2_/Ar flow with a ramping rate of 10 °C /min ranging from 50 °C to 800 °C. A thermal conductivity detector (TCD) was used to record the hydrogen consumption, which was calibrated using a 99.99% CuO reference. Ni K-edge X-ray absorption near edge structure (XANES) investigations were carried out using XAFCA beamline in the transmission mode (Shiga, Japan), and Ni K-edge spectra of pre-reduced catalysts (reduced at 700 °C for 1 h under 10% H_2_/He mixed gases and cooled down to 30 °C under He) were calibrated with respect to the spectrum of a Ni foil and NiO references. Weight loss curves were obtained on a TGA Discovery SDT-650 instrument to evaluate the carbon deposition amount for the spent samples.

### 2.4. Catalytic Activity Testing

The dry reforming of methane (DRM) reaction was performed in a fixed-bed quartz reactor (ID: 7 mm) at atmospheric pressure. Specifically, catalysts (0.15 g, 40–60 mesh) were added to the central reactor tube under the support of quartz wool. The catalyst was pretreated in 10 vol% H_2_/ He (50 mL/min) at 700 °C for 1 h. Then, the reaction gas (CO_2_/CH_4_/Ar/N_2_ = 3:3:3:1) with a total flow rate of 50 mL/min (GHSV = 20,000 mL·g_cat_^−1^·h ^−1^) was introduced to the reactor. The exhausted products were analyzed via a Gas Chromatograph (GC6890, NYSE: A, Palo Alto, CA, USA) equipped with a TCD detector (an HP-Plot capillary column combined with a Carbon-Plot) online.

Temperature-programmed surface reaction-mass spectrometry (TPSR-MS) was employed to evaluate the catalytic activity. An 0.15 g catalyst was pre-reduced as mentioned above. After that, it was cooled down to 300 °C under helium gas with a flow of 45 mL/min. The TPSR-MS analysis was engaged under the mixture gas (CO_2_:CH_4_:Ar = 6/4/15) at 50 mL/min from 300 °C to 850 °C with a heating rate of 10 °C/min. The changes in concentration for CH_4_, CO_2_, H_2_ and CO gases were recorded on a PerkinElmer mass spectrometer.

## 3. Results

### 3.1. XRD Analysis

From the normal XRD patterns (Figure 1), it can be seen that the location of 2θ is labeled at 22.5, 32.1, 39.4, 46.0, 57.2, 67.3 and 76.5°, which are separately ascribed to the (101), (121), (220), (202), (240), (242), and (204) lattice planes of the LaFeO_3_ perovskite with orthorhombic structure (JCPDS PDF# 37-1493), and the peak position remained unchanged despite introducing an active component of NiO (PtO_x_) [25,26]. Moreover, the diffraction peaks located at 37.2, 43.2 and 62.8 should be ascribed to the (111), (200) and (220) planes of cubic NiO [27], and their intensities vary with the content of LaFeO_3_ (from 30 wt.% to 70 wt.%) and hydrothermal temperature (from 120 to 160 °C). The high concentration of LaFeO_3_ combined with the high hydrothermal temperature improved the dispersion of NiO nanoparticles, as displayed in Figure 1a,b. The crystallite sizes of NiO active nanoparticles calculated by Scherrer’s equation were ca. 13 nm (Appendix A). However, no diffraction peak of platinum oxide could be observed due to the detection limit of X-ray diffraction, as indicated in Figure 1b. According to our previous reports, the grain size of NiO deposited on the CeO_2_ (≤5 wt.%) doped SBA-15-CTA could be reduced to ca. 13 nm, and the obtained catalysts exhibit high DRM catalytic activity [24]. Herein, the optimized LaFeO_3_ doping content on the SBA-15-CTA carriers can also enhance the dispersion of the active phase of NiO and lead to high DRM catalytic performance.

### 3.2. N_2_-BET and Morphology Studies

It is generally accepted that the catalyst’s physicochemical properties, such as specific surface area, microstructure, grains size, redox capability and valence states, are closely related to the catalytic activity [28,29]. Figure 2a–c present the N_2_ adsorption–desorption isotherms of the synthesized NiO(PtO_x_)/yLSBA-T catalysts with y value ranges from 30–70 wt.% displaying a type of IV isotherm with H1-type hysteresis loop under the wide hydrothermal condition, revealing their mesoporous characteristic. In addition, the pore size distribution centered at ca. 7 nm for all the catalysts is in good agreement with the hysteresis loops shape (Figure 2a). However, an increase in hydrothermal temperature will result in larger pore size, as depicted in Figure 2b. It is reported that increasing hydrothermal temperature could lead to thinner walls and could broaden mesostructure pore size to some extent [22]. When it comes to the value of the specific surface area (SSA) for the prepared catalysts, the high LaFeO_3_ content coupled with the high in situ growth hydrothermal temperature will cause a decrease in specific surface area. Typically, the SSA of the NiO/3LSBA-140 catalyst was 314.4 m^2^/g, while it was decreased to 123.7 m^2^/g for the high LaFeO_3_ content doped sample (NiO/7LSBA-140) (Figure 2a). On the other hand, the SSA of the NiO/5LSBA-120 catalyst was 219.5 m^2^/g, but the SSA of NiO/5LSBA-160 was reduced to 180.0 m^2^/g (Figure 2b). However, the descending mechanism of SSA was indeed different. The decrease in SSA was caused by the pore blockage for high LaFeO_3_ content modified catalysts, while the decrease in SSA for hydrothermal temperature should be attributed to the enlarged pore diameter, though a 2D-hexagonal mesostructure (Appendix A) can be formed under a wide range of hydrothermal temperature from 120 to 160 °C.

### 3.3. TEM Measurements

Regarding the introduction of platinum noble metal, it was noted that the pore structure and the value of SSA for the NiOPtO_x_/5LSBA-160 catalyst were almost unchanged compared to the parent catalyst of NiO/5LSBA-160, as shown in Figure 2c and Table 1. Moreover, all the prepared NiO(PtO_x_)/yLSBA-T catalysts showed a rod shape morphology (Figure 2d), and the basic morphology was mainly derived from hydrothermal conditions, which corresponds to our previous reports [22]. In order to investigate the relationship between the catalytic activity and particle size of the catalysts, TEM was employed on the outstanding catalysts, namely reduced NiO/5LSBA-160 (Figure 3a) and NiOPtO_x_/5LSBA-160 (Figure 3b) catalysts. It was inferred that introducing Pt could target a higher dispersion of active Ni nanoparticles than that of single LaFeO_3_-doped Ni/5LSBA-160 catalysts. This is due to the stronger interaction of Ni-Pt [30].

### 3.4. H_2_-TPR for Catalysts

To understand the impact of the high SSA LaFeO_3_ carrier on the reductivity of large amounts of active O species of catalysts under various conditions (hydrothermal temperature, concentration and Pt additive), H_2_-TPR experiments were performed, and the reflected results are shown in Figure 4. It was found that the reduction peak can be effectively lagged by inducing the LaFeO_3_ owing to the strong interaction between NiO and LaFeO_3_; the high LaFeO_3_ was more conducive to the formation of a stronger NiO–support interaction under the same hydrothermal temperature. Two reduction peaks appeared at approximately 415 and 480 °C for NiO/1LSBA-140, while significantly higher positions (460 and 669 °C) were observed for NiO/5LSBA-140. Moreover, the interaction of the NiO–support can be further enhanced via the adjustment of hydrothermal temperature. The peak position of NiO/5LSBA-160 is located at 484 and 644 °C, which should be attributed to the reduction peaks of amorphous NiO species having stronger interaction with the LaFeO_3_ which can be retained as highly dispersed active Ni centers after reduction [23]. However, the reduction peak temperature was significantly reduced to 423 °C after the addition of Pt precious metal, which was assignable to a stronger interaction of Ni-O-Pt than that of the NiO–supports [31].

### 3.5. XANES Spectra for Catalysts

According to the above discussion on the dispersion as well as reducibility of NiO loaded on the high SSA LaFeO_3_-SBA-15-CTA catalysts, it was found that the introduction of high SSA LaFeO_3_-SBA-15-CTA with the optimized content (50 wt.%) under the adjusted hydrothermal environment (160 °C) could enhance the interaction of NiO–supports, and improve the dispersion of Ni active species, leading to better activity and sintering resistance during the DRM reaction. XANES is an effective tool for analyzing the valence state of metal elements [32]. As presented in Figure 5, the metallic state of Ni can be well retained by introducing the optimized content of LaFeO_3_ and alloying of Pt-Ni compared to the XANES spectra of Ni foil and NiO, being consistent with what is studied in the literature [33,34]. It was confirmed that the addition of both LaFeO_3_ and Pt plays a key role in stabilizing the metallic state of active Ni species, most probably through the strong interaction between Ni and high SSA mesoporous LaFeO_3_-silica (LSBA) matrix. However, the Ni-Pt nanoparticles present a stronger interaction than that of the Ni-LaFeO_3_ species due to their intrinsic properties, which are proven by numerous studies [35,36]. Therefore, it was inferred that the alloying of Ni-Pt could enhance their tolerance to carbon coking more than that of a Pt-free Ni/5LSBA-160 catalyst.

### 3.6. Catalytic Activity in DRM Reaction

Figure 6a–c depict the catalytic activity of the series of Ni/yLSBA-140 catalysts. It was found that the most optimized doping content of LaFeO_3_ was listed as 50 wt.%, and corresponding catalysts (Ni/5LSBA-140) exhibited higher catalytic performance and thermal stability (48 h) compared to the undoped 5Ni/SBA-15-CTA catalyst [24], where the CO_2_ conversion (Figure 6a), CH_4_ conversion (Figure 6b), H_2_/CO molar ratio (Appendix A) and H_2_ selectivity (Figure 6c) were increased to 70.4%, 66.0%, 87.5% and 86.5%, respectively. Moreover, their catalytic activity was further promoted by controlling the hydrothermal temperature, as Figure 6d–f displays. Notably, the corresponding activity values were increased to 72.1%, 65.5%, 91.4% and 92.0% over the Ni/5LSBA-160 catalyst. This indicates that the catalytic performance and thermal stability, namely CH_4_ conversion (Figure 6d), CO_2_ conversion (Figure 6e), H_2_/CO molar ratio (Appendix A) and H_2_ selectivity (Figure 6f), was significantly enhanced by LaFeO_3_ addition as well as the adjustment of the hydrothermal conditions.

### 3.7. Temperature Programmed Reaction

It was confirmed that the dispersion or the particle sizes of active Ni nanospecies could determine the catalytic activity of Ni-based catalysts for DRM. The well-dispersion Ni particles lead to high catalytic activity [37]. The introduction of the noble metal of Pt improves the dispersion of nickel due to a stronger interaction of Ni-Pt alloy compared to that of the Ni-LaFeO_3_ [30]. In this contribution, it can be observed that the conversion of CO_2_ (Figure 6g) and CH_4_ (Figure 6h), H_2_/CO molar ratio (Appendix A) and H_2_ selectivity (Figure 6i) over the Pt-decorated Ni/LaFeO_3_-SBA-CTA (PtNi/5LSBA-160) catalyst exhibited higher catalytic activity and maintained better stable catalytic activity than that of the Pt-free Ni/LaFeO_3_-SBA-CTA (Ni/5LSBA-160) sample. Moreover, Temperature Programmed Surface Reactions (TPSR-MS) were performed to illustrate the alloying effects of Pt on catalytic performance, as depicted in Figure 7. On the one hand, it was found that there exists a lower temperature for the DRM reaction being initiated over the Pt-doped 5Ni/5LSBA-160 catalyst with a higher reaction rate than the undoped 5Ni/5LSBA-160 sample from the MS signals of the species detected, suggesting that the alloying of the Ni-Pt/5LSBA-160 catalyst process has a stronger capability for activating CO_2_ and CH_4_ than that of the undoped Ni/5LSBA-160 catalyst. On the other hand, the initial temperature for CO_2_ conversion was obviously lower than that of the CH_4_ conversion over both catalysts, which corresponds well to the catalytic activity. This should mainly be attributed to the reversed water gas shift (RWGS) side reaction (Equation (4)), co-existing with DRM [38]. However, it can be inferred that the alloying Pt-Ni nanoparticles improve their interaction to receive better activity, stability and resistance to carbon deposition.
(4)$${\mathrm{CO}}_{2}+H_{2}\;⇌\mathrm{CO}+H_{2}O$$

### 3.8. TG for Spent Catalysts

TGA analysis was proved as an effective technique to evaluate the tolerance capacity to carbon-coking for DRM catalysts. As displayed in Figure 8a, it was found that the weight loss profiles of the spent catalysts showed an inverted volcanic shape (inserted image of Figure 8a) with the increase of LaFeO_3_ contents up to 70 wt.%. Over the Ni/5LSBA-140 catalyst (spent for 12 h), only 8.1% weight loss was detected, which might be closely associated with the enhanced interaction of Ni-LaFeO_3_. From a practical point of view, having higher activity and thermal stability for catalysts is urgently needed. Figure 8b show the weight loss curves of the spent Ni (Pt)/5LSBA-T (T = 120, 140 and 160 °C) catalysts for 48 h. It can be seen that the weight loss increased with the increase of hydrothermal temperature up to 160 °C and the introduction of noble Pt, which might be related to enhanced catalytic performance. The weight loss for the spent Ni/5LSBA-140 catalyst for 48 h was 10.2%, slightly higher than the spent catalyst for 12 h (8.1%), while the weight loss over the NiPt/5LSBA-160 catalyst was increased to 35.8%. However, the types of carbonaceous species deposited on the spent Pt-doped Ni/5LSBA-160 and Pt-free Ni/5LSBA-160 catalysts are evidently different, as shown in the inserted picture of Figure 8b. The DTA peak (643 °C) for the Pt-doped Ni/5LSBA-160 catalyst was lower than that of the Pt-free Ni/5LSBA-160 sample (655 °C), indicating the carbon deposits on the NiPt/5LSBA-160 catalyst was more reactive and easily removed by oxidation under the DRM process to meet the practical application [39].

## 4. Conclusions

In this contribution, a series of mesoporous perovskites with large SSAs were successfully prepared by the in-situ growth of LaFeO_3_ on mesoporous SBA-15-CTA supports under a hydrothermal process. The optimized content (50 wt.%) of LaFeO_3_ was used as a Ni-based catalyst support to synthesize DRM catalysts of Ni/5LSBA-160. It was found that the SSA of mesoporous 5LaFeO_3_-SBA-15-CTA can be up to 180 m^2^/g, which displays it as an excellent DRM catalysts carrier candidate to obtain enhanced catalytic activity and anti-coking properties due to the controllable interaction of Ni-LaFeO_3_ by adjusting the content of LaFeO_3_ and hydrothermal temperature (e.g., 160 °C). Moreover, adding noble Pt over Ni/5LSBA-160 catalyst would further enhance the interaction between Ni and the support to promote the dispersion of active Ni species and achieve a higher syngas formation rate as the TPSR depicted. In addition, the capacity of anti-coking and resistance to sintering can also be improved compared to the Pt-free Ni/5LSBA-160 catalyst sample. This study will pave the way for designing high SSA mesoporous perovskite and utilizing it in heterogeneous catalysis.

## Figures and Tables

**Figure 1 nanomaterials-12-01451-f001:**
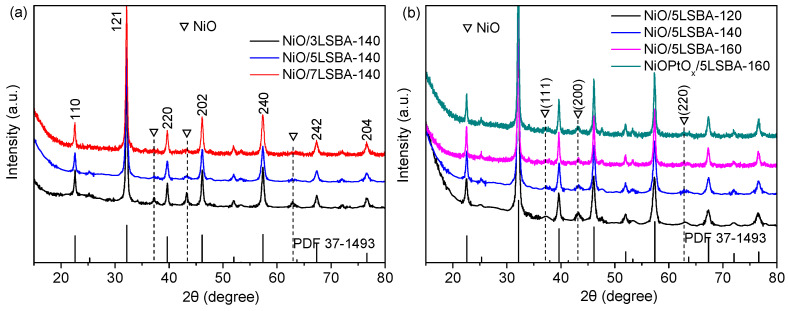
XRD patterns of the NiO(PtO_x_)/yLSBA-T (y means the content of LaFeO_3_, T stands for the reaction temperature) catalysts by varying LaFeO_3_ loading (**a**) and hydrothermal temperature (**b**).

**Figure 2 nanomaterials-12-01451-f002:**
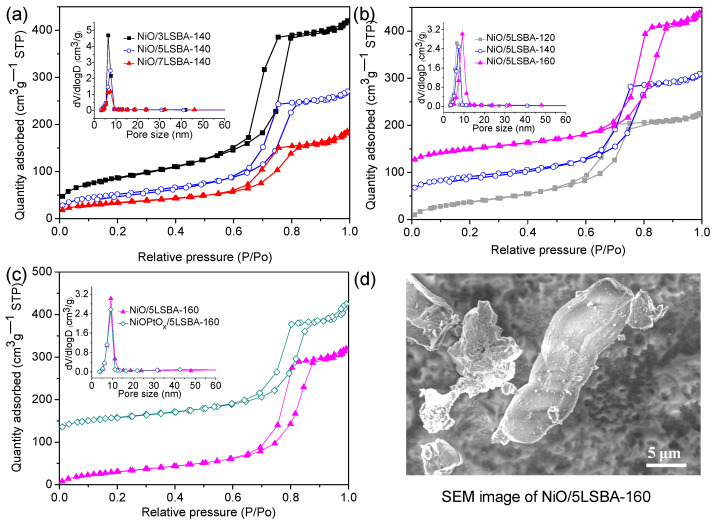
N_2_-BET profiles of the NiO(PtO_x_)/yLSBA-T catalysts by varying LaFeO_3_ loading (**a**), hydrothermal temperature (**b**) and Pt modification (**c**); SEM image of Ni/5LSBA-160 (**d**).

**Figure 3 nanomaterials-12-01451-f003:**
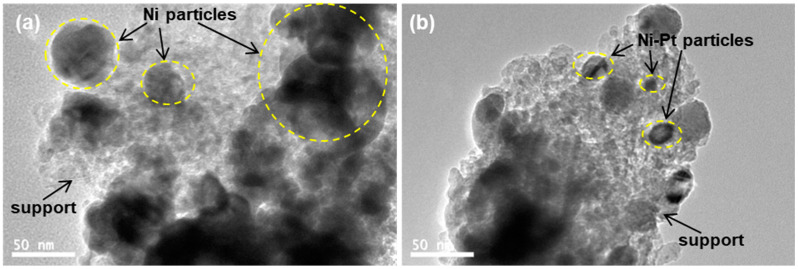
TEM images and nickel particle size of the reduced catalysts: Ni/5LSBA-160 (**a**) and NiPt/5LSBA-160 catalysts (**b**).

**Figure 4 nanomaterials-12-01451-f004:**
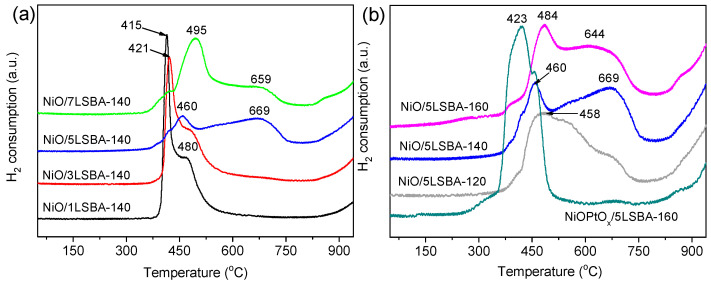
H_2_-TPR curves of the NiO(PtO_x_)/yLSBA-T catalysts by varying LaFeO_3_ loading (**a**), hydrothermal temperature as well as Pt modification (**b**).

**Figure 5 nanomaterials-12-01451-f005:**
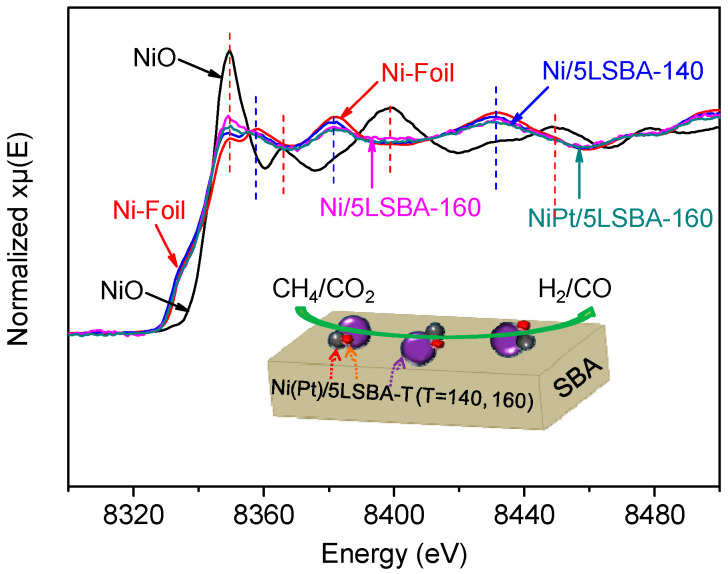
Ni-K-edge XANES spectra of the reduced catalysts and reference standards (NiO and Ni foil).

**Figure 6 nanomaterials-12-01451-f006:**
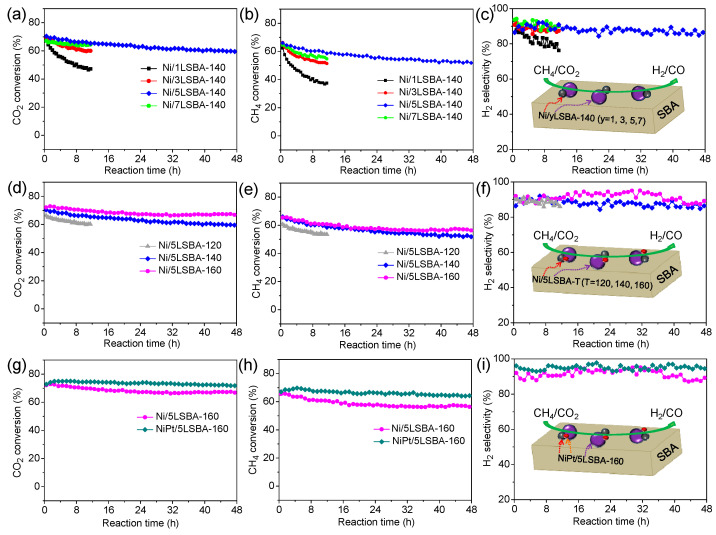
The enhanced catalytic performance on Ni(Pt)/yLSBA-T catalysts by adjusting the LaFeO_3_ loading (**a**–**c**) and hydrothermal temperature (**d**–**f**) as well as the Pt modification (**g**–**i**). (GHSV = 20,000 mL·g_cat_^−1^·h ^−1^).

**Figure 7 nanomaterials-12-01451-f007:**
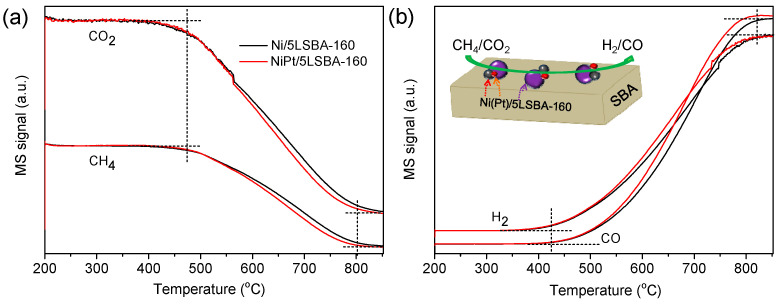
The catalytic activity of 0.5 wt.% Pt co-doped and bare Ni/5LSBA-160 catalysts as a function of temperature (note: the MS signals stand for the content of the detected species), (**a**) stands for the conversion of CO_2_ and CH_4_, (**b**) is the production of H_2_ and CO.

**Figure 8 nanomaterials-12-01451-f008:**
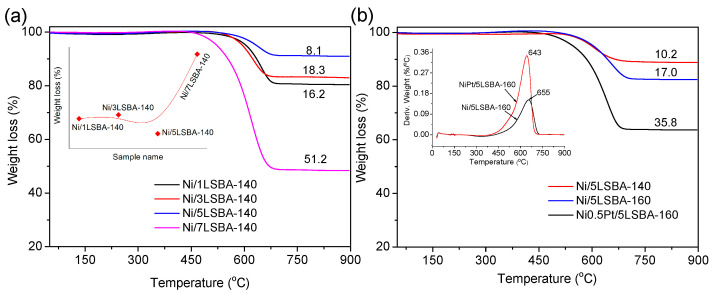
The weight loss profiles of spent Ni(Pt)/yLSBA-T catalysts for 12 h (**a**) and 48 h (**b**), it was noted that the inserted image of (**a**) displayed the weight loss trends for the used catalyst, the inserted image of (**b**) presented the decomposition temperature of carbon species.

**Table 1 nanomaterials-12-01451-t001:** Preparation conditions and physicochemical properties of the synthesized hierarchical structure silica.

Sample Name	LaFeO_3_ Content (wt.%)	Hydrothermal Temperature (°C)	S_BET_ (m^2^/g)	Volume (cm^3^/g)	Pore Size (nm)
NiO/3LSBA-140	30	140	314	0.6	8.2
NiO/5LSBA-140	50	140	187	0.4	8.7
NiO/7LSBA-140	70	140	124	0.3	9.1
NiO/5LSBA-120	50	120	219	0.4	6.8
NiO/5LSBA-160	50	160	180	0.5	11.7
NiOPtO_x_/5LSBA-160	50	160	176	0.5	11.4

## Data Availability

The data presented in this study are available from the corresponding authors upon reasonable request.

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
