# Peer review of "Preparation of Novel Mesoporous LaFeO3-SBA-15-CTA Support for Syngas Formation of Dry Reforming"

_nanomaterials, 2022, doi:10.3390/nano12091451_

Round 1
Reviewer 1 Report
This is an interesting paper on the preparation of mesoporous LaFeO3-SBA-15-CTA doped with Ni and Pt and their use for the syngas formation of dry reforming. I think the paper can be published in Nanomaterials after the authors take into consideration the following points:
- Section 3.1. (XRD analysis): I think the authors should state which is the NiO lattice plane detected at approximately 38 degrees
- The authors on page 4 - line 167 state that the size of NiO particles (based on Scherer’s equation) is c.a. 13 nm. However, looking at Figures 2(a) and (b) there is significant broadening of the NiO reflection for various samples. I think the authors should include a table (possibly in the supplementary information) where they provide the NiO particle size for each catalyst presented.
- Table 1. Looking at the significant figures reported for the Surface areas, Volumes and Pore sizes makes me wonder what is the error in these measurements. I think the author should reconsider the significant figures they have used to report these values and to add the relevant errors.
- Figure 2(d). The authors report the formation of rod like shapes. However, different shapes can be seen in the figure. I wonder if the authors have a larger area image which shows more clearly the rod like shapes (and possibly other shapes formed).
- Figure 3 and relevant discussion in section 3.3. The authors report the formation Ni and Ni-Pt clusters. It is not clear to me how they identify the of Ni-Pt clusters. I think EDS would be necessary to identify the presence of bimetallic particles. Moreover, looking at the XRD data presented earlier in the paper I don’t think there is a shift of the Ni reflections for the NiPt/5LSBA-160 catalyst which would indicate the formation of a bimetallic nanoparticle face. I think the authors should check their discussion and modify the paper accordingly. In my view a technique like EDS or XPS would be necessary to check the presence of a bimetallic phase.
- TEM and SEM images (Figure 2(d) and Figure 3). The SEM shows the formation of crystallites that the authors describe as “rod” shapes. The TEM shows the formation of metal particles. Are the particles observed in the TEM images part of the “rod” like shapes observed in SEM. The TEM particles appear to be on top of an amorphous phase. I think the authors should discuss this point.
- Figure 5 (XANES spectra) and relevant discission on page 7 (lines 249-250): the authors write that the Pt plays a key role in stabilizing the metallic state of the Ni active species. I think, however, that the XANES spectra of the Ni/5LSBA-160 and NiPt/5LSBA-160 are very similar to each other. I think the authors should discuss this or explain at least what are the differences between the two spectra.
- Section 3.6. It would be useful if the authors could report the Turnover frequencies for the various catalysts.
Author Response
Comment 1
nanomaterials-1671933
Title: Preparation of novel mesoporous LaFeO3-SBA-15-CTA support for syngas formation of dry reforming
Dear Reviewer 1,
We would like to express our thanks to you for spending time to handle this article. The corrections are carefully performed and major revision is supplied according to your remarks, and they are highlighted in yellow in the revised version. Thank you.
Best Regards.
Sincerely yours,
Dr. Luming Li,
Email liluming@cdu.edu.cn.
Tel: +86-16608096866.
Comment
This is an interesting paper on the preparation of mesoporous LaFeO3-SBA-15-CTA doped with Ni and Pt and their use for the syngas formation of dry reforming. I think the paper can be published in Nanomaterials after the authors take into consideration the following points:
Response:Thanks for your supporting comments. The detailed response letter will be showed as follows:
- Section 3.1. (XRD analysis): I think the authors should state which is the NiO lattice plane detected at approximately 38 degree.
Response 1: Thanks for your good comment. We have stated NiO lattice plane detected at approximately 38 degrees, as showed in this sentence “Moreover, the diffraction peaks located at 37.2, 43.2 and 62.8 should be ascribed to the (111), (200) and (220) planes of cubic NiO)”. By the way, we also modified the XRD pictures (Figure R1). (Page 4, Line 162-164). Thanks.
Figure R1. XRD patterns of the NiO(PtOx)/yLSBA-T (y means the content of LaFeO3, T stands for the reaction temperature) catalysts by varying LaFeO3 loading (a) and hydrothermal temperature (b).
- The authors on page 4 - line 167 state that the size of NiO particles (based on Scherer’s equation) is c.a. 13 nm. However, looking at Figures 2(a) and (b) there is significant broadening of the NiO reflection for various samples. I think the authors should include a table (possibly in the supplementary information) where they provide the NiO particle size for each catalyst presented.
Response 2: Thanks for your comment. We totally agree that the peak of NiO was broadened by adjusting the content of LaFeO3 and the hydrothermal temperature, indicating the improved dispersion of NiO, however, the degree of improvement is not obvious, as the table R1, showed. Which has been supplied in supplementary information (table R1). (Page 4, line 168). Thanks.
Table R1. Preparation conditions and grain sizes of the synthesized catalysts.
|
Sample name. |
LaFeO3 content (wt.%) |
Hydrothermal temperature (oC) |
Particle size of NiO (nm) |
|
|
NiO/3LSBA-140 |
30 |
140 |
19.2 |
|
|
NiO/5LSBA-140 |
50 |
140 |
14.3 |
|
|
NiO/7LSBA-140 |
70 |
140 |
12.8 |
|
|
NiO/5LSBA-120 |
50 |
120 |
15.3 |
|
|
NiO/5LSBA-160 |
50 |
160 |
13.2 |
|
|
NiOPtOx/5LSBA-160 |
50 |
160 |
13.0 |
|
- Table 1. Looking at the significant figures reported for the Surface areas, Volumes and Pore sizes makes me wonder what is the error in these measurements. I think the author should reconsider the significant figures they have used to report these values and to add the relevant errors.
Response 3: Thanks for your kind reminder, we have checked the significant figures reported for the Surface areas, Volumes and Pore sizes, and made some correction (Table R2). By the way, a Micromeritics ASAP 2420 automatic analyzer was employed to achieve the specific surface area, pore volume and pore sizes, the reproducibility of value has been confirmed and should be trustworthy. Thanks.
Table R2. Preparation conditions and physicochemical properties of the synthesized hierarchical structure silica.
|
Sample name. |
LaFeO3 content (wt.%) |
Hydrothermal temperature (oC) |
SBET (m2/g) |
Volume (cm3/g) |
Pore size (nm) |
|
|
NiO/3LSBA-140 |
30 |
140 |
314 |
0.6 |
8.2 |
|
|
NiO/5LSBA-140 |
50 |
140 |
187 |
0.4 |
8.7 |
|
|
NiO/7LSBA-140 |
70 |
140 |
124 |
0.3 |
9.1 |
|
|
NiO/5LSBA-120 |
50 |
120 |
219 |
0.4 |
6.8 |
|
|
NiO/5LSBA-160 |
50 |
160 |
180 |
0.5 |
11.7 |
|
|
NiOPtOx/5LSBA-160 |
50 |
160 |
176 |
0.5 |
11.4 |
|
- Figure 2(d). The authors report the formation of rod like shapes. However, different shapes can be seen in the I wonder if the authors have a larger area image which shows more clearly the rod like shapes (and possibly other shapes formed).
Response 4: It is an interesting question. The rod like shape was easily formed under this hydrothermal condition, which has been reported in our previous study (Improved facile synthesis of mesoporous SBA-15-CTA using citric acid under mild conditions. Journal of Solid State Chemistry, 2020, 282: 121079.). Moreover, a larger area image was also offered in Figure R2. Thanks.
Figure R2. SEM images of 5NiO/5LSBA-160 sample.
- Figure 3 and relevant discussion in section 3.3. The authors report the formation Ni and Ni-Pt clusters. It is not clear to me how they identify the of Ni-Pt clusters. I think EDS would be necessary to identify the presence of bimetallic particles. Moreover, looking at the XRD data presented earlier in the paper I don’t think there is a shift of the Ni reflections for the NiPt/5LSBA-160 catalyst which would indicate the formation of a bimetallic nanoparticle face. I think the authors should check their discussion and modify the paper accordingly. In my view a technique like EDS or XPS would be necessary to check the presence of a bimetallic phase.
Response 5: Thanks for your valuable comment. We have corrected the discussion that Ni and Ni-Pt nanoparticles were formed and the Pt could target a higher dispersion of active Ni nanoparticles than that of single LaFeO3 doped Ni/5LSBA-160 catalysts, this is due to stronger interaction of Ni-Pt. By the way, we have made EDS analysis (Figure R3) on 5NiOPtOx/5LSBA-160 catalyst, and the Ni and Pt are always dispersed evenly. On the hand, the Ni-Pt nanoparticles would present a stronger interaction than that of Ni-LaFeO3 species due to their intrinsic properties, which has been proven by numerous studies (such as Transactions of Tianjin University 2019, Novel Pt–Ni Bimetallic Catalysts Pt(Ni)–LaFeO3/SiO2 via Lattice Atomic-Confined Reduction for Highly Efficient Isobutane Dehydrogenation; Catalysis Letters, 2019, Perovskite-Derived Pt–Ni/Zn(Ni)TiO3/SiO2 Catalyst for Propane Dehydrogenation to Propene, and et al. ) . Therefore, it was inferred that the alloying of Ni-Pt can be well formed by co-impregnation method in this contribution, and alloying of Ni-Pt are abundant existed everywhere. (Page 8 Line 214). Thanks.
Figure R3. EDS images of 5NiOPtOx/5LSBA-160 catalyst.
- TEM and SEM images (Figure 2(d) and Figure 3). The SEM shows the formation of crystallites that the authors describe as “rod” shapes. The TEM shows the formation of metal particles. Are the particles observed in the TEM images part of the “rod” like shapes observed in SEM. The TEM particles appear to be on top of an amorphous phase. I think the authors should discuss this point.
Response R6: It is an interesting comment. Amorphous species may also have certain morphologies, and they also show some certain lattice fringes. For example, SBA-15, reported 2D mesoporous materials by Prof. Dongyuan Zhao, always presents rod like shape and certain lattice fringes, as the literature displayed in Figure R4 (Science 279, 548 (1998)), because the lattice fringes only reflect the periodic structure. Thanks.
Figure R4. The SEM and HRTEM images of SBA-15 (literature)
- Figure 5 (XANES spectra) and relevant discission on page 7 (lines 249-250): the authors write that the Pt plays a key role in stabilizing the metallic state of the Ni active species. I think, however, that the XANES spectra of the Ni/5LSBA-160 and NiPt/5LSBA-160 are very similar to each other. I think the authors should discuss this or explain at least what are the differences between the two spectra.
Response R7: It is a good suggestion, EXAFS analysis was not performed in this paper, and bond length and coordination of Ni and Pt would be displayed in near future. However, the TEM, H2-TPR show the promotion effect of Pt on the improvement of the dispersion and reduction property of NiO, and the result of catalytic activity also present the its enhanced tolerance to carbon coking than that of Pt-free Ni/5LSBA-160 catalyst sample, because the Ni-Pt nanoparticles would present a stronger interaction than that of Ni-LaFeO3 species due to their intrinsic properties. (Page 7, line 241-256) Thanks
- Section 3.6. It would be useful if the authors could report the Turnover frequencies for the various catalysts.
Response R8: It is a good comment. In this contribution, one-step in situ-growth hydrothermal process was firstly employed to effectively prepare a series of LaFeO3/SBA-15-CTA hybrid materials with large specific surface area, and these mesoporous perovskites have been used as the carriers to synthesize Ni(Pt)/LaFeO3-SBA hybrid catalysts for DRM reaction. The obtained composites catalysts displayed a higher activity and resistance to carbon deposition for dry reforming process. This finding uncovers a promising prospect for high SSA mesoporous perovskite preparation and utilizations in dry reforming. Therefore, the novelty for this paper presents the preparation of perovskite with large specific surface area, and the catalytic activity was only one model application and could be weakly displayed. The turnover frequencies for the various catalysts would be deeply investigated in near future, thanks for your good suggestions.

Reviewer 2 Report
The manuscript titled “Preparation of novel mesoporous LaFeO3-SBA-15-CTA support for syngas formation of dry reforming” presented by Luming Li, Song Wu, Hongmei Li, Jie Deng, Junshan Li deals with the perovskite-based catalysts for dry reforming of methane. However, there are issues that could be clarified or took into account:
- Line 15: Use of abbreviation “DRM” is not necessary. The “DRM” does not used in the abstract.
- Line 17: Use of term “in-situ” is incorrect. The term sounds “in situ”.
- Line 19: “noble Pt”, do the authors know any unnoble platinum?
- Line 31-33: I recommend authors to calculate the reaction enthalpies for the target temperature of DRM reaction not for 298K.
- Line 42: “enhanced metal-support interaction” is incorrect, the generally accepted form of the effect is “strong metal-support interaction” or SMSI effect.
- Line 47: It is unlikely that catalyst support could demonstrate any activity in the DRM.
- Line 48-49: “many attentions” is wrong, the correct form is “much attention”.
- Line 123: Add the wavelength and use the correct terms “Cu Kalpha1,2 irradiation with wavelength of … nm.” I am not sure that D8 Advance has a monochromator to filter Kalpha2 component. To my best knowledge D8 Advance has only (Z-1) nickel filter.
- The section 2.3 should be extended. For example, resolution of TEM, the beamline where XAS was done, etc.
- TEM study is poor and should be redone. First, the observed particles are not clusters but nanoparticles with size in range of 15-50 nm and more. TEM should be done with scale of 10 nm (high resolution). EDX mapping is demanded. The interplanar distances should be measured for nickel nanoparticles.
The choice of methods to study the catalysts looks well. But the TEM could reveal more information about the structure of catalysts. I recommend the authors redone TEM study more carefully.
Author Response
Comment 2
nanomaterials-1671933
Title: Preparation of novel mesoporous LaFeO3-SBA-15-CTA support for syngas formation of dry reforming
Dear Reviewer2,
We would like to express our thanks to you for spending time to handle this article. The corrections are carefully performed and major revision is supplied according to your remarks, and they are highlighted in yellow in the revised version. Thank you.
Best Regards.
Sincerely yours,
Dr. Luming Li,
Email liluming@cdu.edu.cn.
Tel: +86-16608096866.
Comment
The manuscript titled “Preparation of novel mesoporous LaFeO3-SBA-15-CTA support for syngas formation of dry reforming” presented by Luming Li, Song Wu, Hongmei Li, Jie Deng, Junshan Li deals with the perovskite-based catalysts for dry reforming of methane. However, there are issues that could be clarified or took into account:
Response: Thanks for your comment and the detailed response letter was displayed as follows:
- Line 15: Use of abbreviation “DRM” is not necessary. The “DRM” does not used in the abstract.
Response 1: “DRM” has been removed, thanks
- Line 17: Use of term “in-situ” is incorrect. The term sounds “in situ”.
Response 2: “in situ” has been used to replace the “in-situ”, thanks
- Line 19: “noble Pt”, do the authors know any unnoble platinum?
Response 3: Haha, Thanks for your humorous comment, the noble has been deleted. Thanks.
- Line 31-33: I recommend authors to calculate the reaction enthalpies for the target temperature of DRM reaction not for 298K.
Response 4: It is a valuable comment. We totally agree that the reaction enthalpies for the target temperature of DRM reaction should be calculated to state the promotion effect of catalyst. However, in this contribution, one-step in situ-growth hydrothermal process was firstly employed to effectively prepare a series of LaFeO3/SBA-15-CTA hybrid materials with large specific surface area, and these mesoporous perovskites have been used as the carriers to synthesize Ni(Pt)/LaFeO3-SBA hybrid catalysts for DRM reaction. The obtained composites catalysts displayed a higher activity and resistance to carbon deposition for dry reforming process. This finding uncovers a promising prospect for high SSA mesoporous perovskite preparation and utilizations in dry reforming. Therefore, the novelty for this paper presents the preparation of perovskite with large specific surface area, and the catalytic activity was only one model application and the catalytic kinetics could be weakly displayed. The reaction enthalpies would be deeply investigated in near future, Thanks
- Line 42: “enhanced metal-support interaction” is incorrect, the generally accepted form of the effect is “strong metal-support interaction” or SMSI effect.
Response 5: We have revised “enhanced metal-support interaction” into “strong metal-support interaction”. Thanks
- Line 47: It is unlikely that catalyst support could demonstrate any activity in the DRM.
Response 6: Thanks for your comment. We are sorry for the unclear elaboration, and we have rewritten this sentence. “Up to now, various materials such as Al2O3, ZrO2, MgO, CeO2, mesoporous silica, and molecular sieve have been used as catalysts support or additives and can enhance the catalytic activity of active species such as Ni particles in DRM”
- Line 48-49: “many attentions” is wrong, the correct form is “much attention”.
Response 7: Thanks, we have corrected it.
- Line 123: Add the wavelength and use the correct terms “Cu Kalpha1,2 irradiation with wavelength of … nm.” I am not sure that D8 Advance has a monochromator to filter Kalpha2 component. To my best knowledge D8 Advance has only (Z-1) nickel filter.
Response 8: Thanks for your kind reminder, we have added the wavelength (0.15418 nm) of X-ray. By the way, D8 D8 ADVANCE has Cu Kα1. Page 3 line 123.
- The section 2.3 should be extended. For example, resolution of TEM, the beamline where XAS was done, etc.
Response9: Thanks for your comment. The section 2.3 has been extended. X-ray diffraction (XRD, 2θ = 5–80o) and small-angle XRD (2θ = 0–6o) were carried out on a X-ray diffractometer (Bruker D8 Advance) using Cu Kα1 irradiation (λ=0.15418 nm). N2 adsorption-desorption experiments were performed on a Micromeritics ASAP 2420 automatic analyzer to achieve the specific surface area, pore volume. Prior to testing, the samples were pretreated at 200 oC for 12h to remove impurities. The microstructure of samples was analyzed via a scanning electron microscopy (SEM, JEOL, JEM-2100F) and transmission electron microscopy (TEM, FEI Tecnai G2 F20) at 200 kV. An AutoChem II 2920 (Thermo Scientific) was engaged to study the reduction behavior of the NiO(PtOx)/yLSBA-T catalysts. In a typical experiment, 30 mg sample was used, which was pretreated under Ar flow (30 mL/min) at 300 oC for 1h. After that, the sample was heated in a 30 mL/min 10% H2/Ar flow with a ramping rate of 10 oC /min ranges from 50 oC to 800 oC. A thermal conductivity detector (TCD) was used to record the hydrogen consumption, which was calibrated by using a 99.99% CuO reference. Ni K-edge X-ray absorption near edge structure (XANES) investigations were carried out using XAFCA beamline in the transmission mode, and Ni K-edge spectra of pre-reduced catalysts (reduced at 700 oC for 1 h under 10%H2/He mixed gases and cooled down to 30 oC under He) were calibrated with respect to the spectrum of a Ni foil and NiO references. Weight loss curves were obtained on a TGA Discovery SDT-650 instrument to evaluate the carbon deposition amount for the spent samples. Page 3. Line 121-139. Thanks.
- TEM study is poor and should be redone. First, the observed particles are not clusters but nanoparticles with size in range of 15-50 nm and more. TEM should be done with scale of 10 nm (high resolution). EDX mapping is demanded. The interplanar distances should be measured for nickel nanoparticles. The choice of methods to study the catalysts looks well. But the TEM could reveal more information about the structure of catalysts. I recommend the authors redone TEM study more carefully.
Response 10: Thanks for your valuable comment. We are sorry for our mistake, the TEM do present the nanoparticles, not clusters, and we have corrected it in line 216. We totally agree that nanoparticles with size in range of 15-50 nm and more, and are uniformly distributed, though both the high hydrothermal temperature and Pt additive were introduced into synthesis process from the TEM images. As for the resolution of TEM, it's really hard to achieve the high-quality TEM picture, because the Ni nanoparticles possess magnetic property, and the screen wobbles a lot at the high resolution such at 10 nm, which has been redone several times. Moreover, the amorphous support of silica would also produce some bubbles. However, some interesting phenomena have been observed, and will be deeply studied in the near future. Thanks for your supporting.

Round 2
Reviewer 1 Report
I think the authors addressed nicely the comments of the referees. I think the manuscript can be published without further revision.